# An Assessment of Landscape and Land Use/Cover Change and Its Implications for Sustainable Landscape Management in the Chittagong Hill Tracts, Bangladesh

Masheli Chakma [1,†], Umer Hayat [2,†] , Jinghui Meng [1,*] and Mohammed A Hassan [3]

1 Research Center of Forest Management Engineering of National Forestry and Grassland Administration, Beijing Forestry University, Beijing 100083, China; mashelichakma@yahoo.com
2 Sino-France Joint Laboratory for Invasive Forest Pests in Eurasia, Department of Forest Protection, School of Forestry, Beijing Forestry University, Beijing 100083, China; oomarcassi6116@gmail.com
3 Department of Forestry Sciences, University of Zalingei, Zalingei P.O. Box 6, Sudan
* Correspondence: jmeng@bjfu.edu.cn
† These authors contributed equally to this work.

**Abstract:** Human-caused environmental change has profoundly impacted resource management and land use patterns in Bangladesh's Chittagong Hill Tracts. This study used multi-temporal Landsat images from 1998, 2008, and 2018 to analyze land use and land cover changes, particularly those associated with forest cover changes, in Bangladesh's Chittagong Hill Tracts. Using object-based image classification, Landsat images from 1998, 2008, and 2018 were separated into four categories based on their dominant land use and land cover features: forest, grassland, water bodies, and bare land. Post-classification comparison was used to assess the degree and frequency of change, and this method was further developed to evaluate the balance, fluctuation, and adaptation of forests. In addition, the spatial structure of land cover and temporal trajectories related to changes in forest cover were studied. The CA–Markov chain model was also used to anticipate the 2048 LULC map. The image classification of the years 1998, 2008, and 2018 showed that the overall accuracy was 89.65%, 84.44%, and 86.26%; producer accuracy was 90.00%, 68.75%, and 72.22%; and the Kappa coefficient was 85.68, 82.84, and 76.36, respectively. The results showed that between 1998 and 2018, forest cover increased by 58.03%, transforming grassland to forest; grassland increased by 29.50%, converting bare land to grassland; and forest conversion to grassland was 13.34%. In addition, the result of the landscape metric revealed that during the whole study period, class level indicated a fragmentation of forest, bare land, grassland, and water in the CHT, and landscape level indicated by Shannon's Diversity Index and Shannon's Evenness Index showed a slight decrease in the land. Based on the CA–Markov model, forest area is predicted to expand to 9129 Km$^2$ in 2048; however, other land uses (bare land and grassland) continue to decrease. This substantial increase in forest cover results from effective forest management based on community forestry practices and the successful execution of Bangladesh's national forest strategy. However, as Bangladesh's population rises, so does the country's need for lumber/timber. Bangladesh's government should revise its forest policy to meet the local community's needs without endangering the forest, and policymakers must take climate change seriously. Our strategy for evaluating the critical indicators of changes in forest cover and pathways of change will aid in connecting these patterns to the dynamics of change, such as deforestation and reforestation. It would therefore serve as a framework for developing effective conservation and management plans for the Chittagong Hill regions in Bangladesh.

**Keywords:** remote sensing; GIS; LULC; dynamic change detection; human driving force; community forestry; forest management

## 1. Introduction

The recognition of the invaluable services provided by tropical forests to both host nations and the global community is widely acknowledged [1]. These unique ecosystems, characterized by their unparalleled array of plant and animal species, possess irreplaceable biodiversity and genetic resources [2,3]. Since the Earth Summit, forest policies in numerous countries have prioritized the objective of sustainable forest management, regardless of the extent of human interventions within forested areas [4]. Forests and woodlands are vital to keeping the environment carbon-free [5–7]. However, ecological systems, such as forests, experience constant changes due to natural biological processes, resulting in continued instability [4].

Bangladesh has approximately 2.52 million hectares of tropical forest, accounting for 10% of its area [8]. Rural areas house over two-thirds of Bangladesh's population, and their livelihoods are closely linked to the forests, either directly or indirectly [9]. For several decades, the national forests in Bangladesh have faced significant and rapid depletion, reaching a critical stage of concern [1]. The natural forests of Bangladesh experienced a consistent annual decline of 2.1% over 20 years until the early 1980s, which further accelerated to a rate of 2.7% between 1984 and 1990 [10]. Between 1964 and 1985, there was a decline in the growing stock within Chittagong Hill Tracts reserve forests, decreasing from 23.8 million $m^3$ to below 19.8 million $m^3$ [11]. Between 2000 and 2005, around 2000 hectares of forest cover were lost annually [12].

As a result of degradation, forest communities are motivated to utilize their traditional knowledge and practices to engage in activities such as conservation, reforestation, bushfire control, and the prevention of illegal forest exploitation and encroachment [13]. The combined efforts of these local communities have led to the regeneration of forested lands and the enhancement of biodiversity levels [14]. Several policies and practices in South and Southeast Asia have emerged centered around community-based forest management [15,16]. The global community places significant emphasis on preserving biodiversity, promoting forest health, ensuring sufficient forest productivity, and safeguarding the socio-economic functions associated with forest resources [17].

In Bangladesh, traditional forest management techniques have historically aimed to achieve economic and ecological objectives [18]. However, rapid deforestation occurred due to various socio-economic and socio-political factors [19,20], diminishing the effectiveness of traditional forest planning and management approaches. Unplanned human activities and unforeseen pressures exceeded planned conservation efforts, resulting in extensive deforestation and fragmentation of forest resources [19]. Given the country's dense population and limited land area, policymakers had to explore alternative management practices. In the late 1970s, social forestry was introduced as a successful alternative that transitioned the Forest Department's role from a custodial to a participatory model, involving local communities in forest protection, reforestation activities, and benefit-sharing arrangements [21].

During Bangladesh's current period of sovereignty, the Forest Act underwent its first amendment in 1989, which aimed to enhance forest protection by imposing stricter penalties and limiting the discretionary powers of forest officials and local magistrates [18]. Although this amendment primarily focused on strengthening traditional forest protection measures, it did not introduce the concept of social forestry until 2000, when another amendment was introduced, leading to the emergence of social forestry in Bangladesh [22]. The Forest (Amendment) Act of 2000 marked a significant milestone as it paved the way for the formulation of the groundbreaking 2004 Social Forestry Rules (SFR) by the government [18]. Bangladesh's fundamental principle of social forestry revolves around integrating local communities in reforestation activities, aiming to achieve multiple objectives encompassing ecological, economic, and social benefits [23].

One of the main requirements for global change research is to evaluate and track the condition of the earth's surface [24,25]. As the foundation for all living things and a key factor in global climate change, vegetation classification and mapping are crucial tech-

nological undertakings for managing natural resources [25,26]. Land use/cover change (LUCC) is most frequently associated with logging, globalization, and agricultural expansion that alter the natural vegetation [27,28]. At both the local and global levels, LUCC causes several environmental issues, such as biodiversity loss brought on by greenhouse gas emissions [28,29], variations in land surface temperature (LST), and shifts in precipitation [30]. Urbanization's adverse environmental effects, which include population increase, extensive infrastructure development, and constantly shifting landscapes, are a worldwide issue [31].

Bangladesh, known for its high population density, faces increasing land pressure, particularly in forested areas, due to food production, urban settlements, and industrial development [1]. Approximately 60% of the total land is utilized for agriculture, which serves as a primary source of livelihood for over two-thirds of the rural population [32]. However, limited land availability per person, constrained by geographical factors and inadequate farming practices, hinders the country's food production capacity [9]. The forests and agricultural lands play a crucial role in the lives of people residing in Bangladesh's Chittagong Hill Tracts (CHT) region [33]. In the past, these forested landscapes provided various local and regional benefits, including food, energy, timber, water, and healthcare, while also contributing to national revenue generation. Nonetheless, the exploitation and degradation of forests, which started in the previous century [34] and continue to this day, have significant implications for the sustainable livelihoods of forest-dependent communities in terms of both direct and indirect resources [35].

Bangladesh is not exceptional in these environmental changes. Urbanization has also affected the local environment to certain degrees and made it more susceptible to land degradation. Thus, timely and accurate information about local spatial coverage, distributions of LULC categories, and their dynamics are prerequisites for the country's planning, socio-economic development, and sustainable land management. No evidence-based studies have been conducted in the Chittagong region to understand how implementing a community-based forestry policy affects forest cover change and regional land dynamics. This is the first study of its kind to examine the impact of community-based forestry on land use and land cover change in the Chittagong Hill Tracks and whether or not there are factors that influence this change. This study will highlight the aspects responsible for a land cover change in the Chittagong Hill Tracks. The study aimed to access the dynamic LULC change detection in the Chittagong Hill Tracks using remote sensing data and Geographic Information Systems technologies for 1998, 2008, and 2018. The study also used the cellular automata–Markov model (CA–Markov) to predict future land use changes under a simulated 2048 scenario. The main objectives of this study were: (i) to detect and identify the dynamics change in LULC from 1998 to 2018 in the Chittagong Hill Tracts, Bangladesh; (ii) to identify the driving forces behind these changes; and (iii) to predict the LULC map for the year 2048 using CA–Markov.

## 2. Materials and Methods

### 2.1. Study Area

The Chittagong Hill Tracts are part of the Chittagong Division, located in the southeastern hilly area of Bangladesh, and play a vital role (Figure 1) [9]. It geographically lies between 21.025′ N to 23.045′ N latitude and 91.045′ E to 92.050′ E longitude and covers an area of approximately 13,198.16 Km$^2$, almost 40% of the total forest land area, and 8% of the total land area in Bangladesh [36]. The study area borders Myanmar in the southeast, the Indian state of Tripura to the north, Mizoram to the east, and the Chittagong district in the west. The Chittagong Hill Tracts are divided into three hill districts: Khagrachhari, Bandarban, and Rangamati. The area comprises forest, grassland, watershed, bare land, and agricultural land [9]. Paths and rows cover this area: 136/44, 136/45, and 135/45. In this region, almost twelve ethnic groups are the principal inhabitants, highly dependent on natural resources for their livelihood [37]. The major sources of income are agriculture, livestock, and the harvesting of forest products. The landscape and local people are unique

parts of the Chittagong Hill Tracts. The traditional landscape of Chittagong Hill Tracts combines a mix of land uses: natural forests and plantation forests cover more than 70% of the land area in three districts. The primary plant species in the region are tropical wet evergreen/semi-evergreen and deciduous, classified as 'hill forests'. The large area covers mixed natural and planted forests [34]. This area is characterized by a tropical monsoon climate with three dominant seasons: the dry winter season (November to March), the pre-monsoon season (April to May), and the monsoon season (June to October). The rainfall usually starts to increase from March to June. Annual rainfall ranges from 2540 mm in the north to 2450 mm in the east, reaching nearly 3810 mm in the south and west.

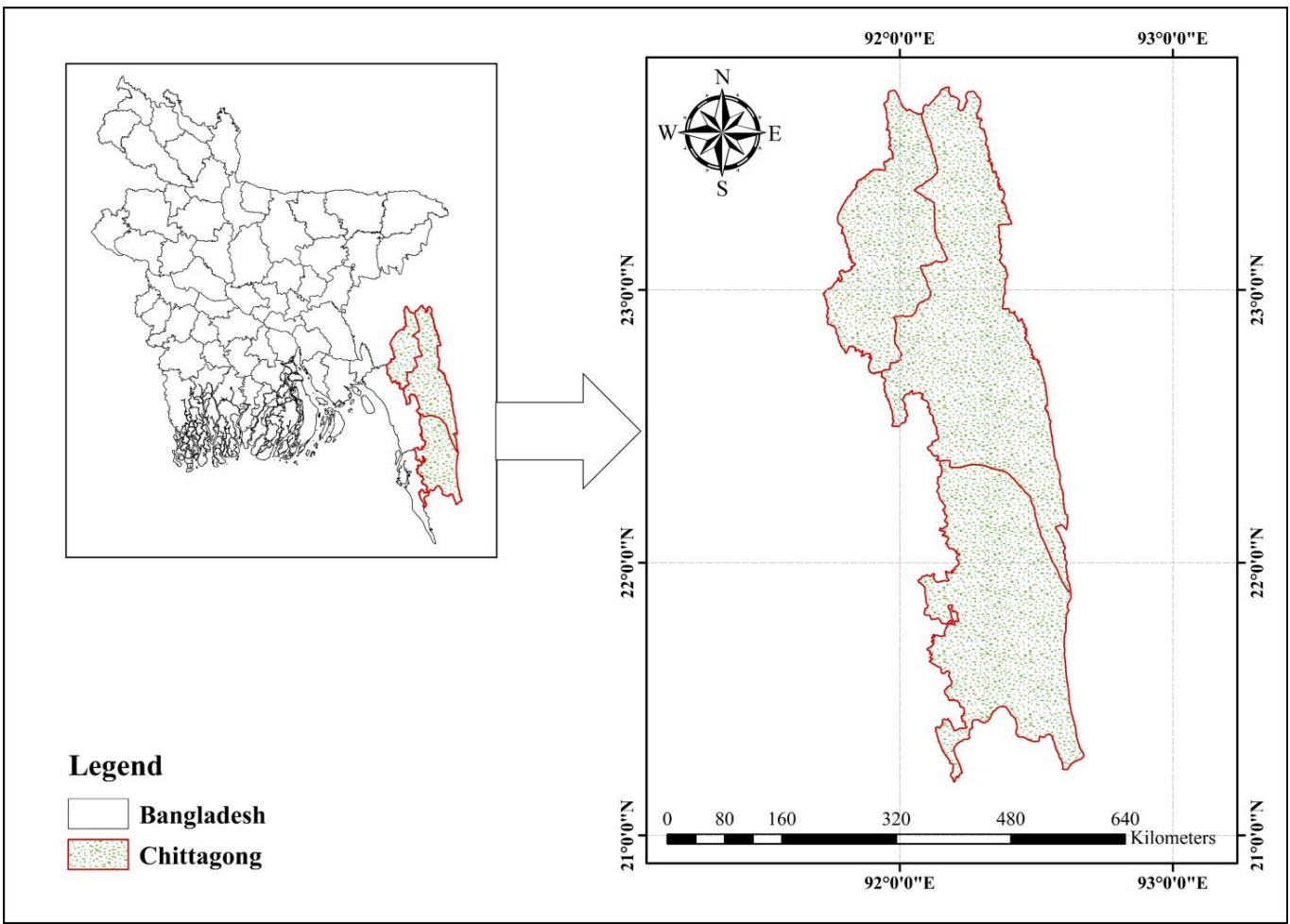

**Figure 1.** Location map of Chittagong Hill Tracts showing the study area in Bangladesh.

*2.2. Remote Sensing Data*

The data used in this study were obtained from various Landsat sensors, including Thematic Mapper (TM), Enhanced Thematic Mapper Plus (ETM+), and Landsat 8 OLI for 1998, 2008, and 2018 (Table 1). A total of 3 senses were acquired between November and December. All images were obtained from the United States Geological Survey (USGS—https://earthexplorer.usgs.gov/ (accessed on 12 January 2023)). All raw satellite images had a cloud cover of less than 10%. Each image cloud was removed by supervised classification extraction by the mask tool in ENVI 5.1. For the Thematic Mapper (TM), the Enhanced Thematic Mapper Plus (ETM+) image band combination was red-green-blue (RGB) [4-3-2], and the Landsat 8 OLI image band combination was red-green-blue (RGB) [5-3-2].

**Table 1.** Characteristics of Landsat data used in the study.

| Years | Satellite | Sensor Type | Spatial Resolution | Date of Acquisition | Image |
|---|---|---|---|---|---|
| 1998 | Landsat 5 | TM | 30 m | December 1998 | GeoTiff |
| 2008 | Landsat 7 | ETM+ | 30 m | November 2008 | GeoTiff |
| 2018 | Landsat 8 | OLI | 30 m | November 2018 | GeoTiff |

*2.3. Satellite Image Pre-Processing*

Geometric rectification is essential for producing corrected LULC maps (Figure 2). For change detection, various pre-processing requirements, for example, geometric correction, radiometric collaboration, and atmospheric corrections, are the most important to avoid errors in the results. In this study, Landsat Collection Level-1 downloaded images were rectified and corrected—geometrically and topographically. Moreover, one of the more significant preconditions for remote sensing data analysis is an atmospheric correction, which has been done by applying the Model FLAASH (Fast Line-of-Sight Atmospheric Analysis of Spectral Hypercubes) to improve image information by transforming radiance as a sensor into surface reflectance values. Mosaic was applied for the paths and rows: 136/44, 136/45, and 135/45. The image pre-processing was done using ENVI 5.1 and ArcGIS 10.5 software.

*2.4. Image Classification*

Image classification was performed using a supervised classification method based on the Maximum Likelihood Classifier of given classes (Table 2). Training samples were selected with sufficient homogeneity to symbolize the spectral and spatial characteristics of each LULC class and maximize classification accuracy. This step is the most significant element of supervised classification because spectral signatures extracted from training samples will define the overall accuracy of the classification and, therefore, the final LULC map.

**Table 2.** Classification scheme/categories.

| Code | LULC Classes | Description |
|---|---|---|
| 1 | Water | Area covered with ponds, lakes, rivers, and seasonal streams. |
| 2 | Bare land | Areas described with non-vegetative cover, rocks, and some limited settlement areas. |
| 3 | Forest | Areas with natural or artificial woody vegetation and canopy cover more than <10%. |
| 4 | Grassland | Areas covered by grasses, agriculture, and some sparse shrubs in between. |

*2.5. Accuracy Assessment*

Accuracy assessment is a post-classification step accomplished through the correspondence evaluation of classified LULC maps against assumed factual geographical reference data [38]. The image classification accuracy was assessed with ground truth data in the form of reference data points obtained from Google Earth images of 1998, 2008, and 2018, used to obtain overall accuracy, user accuracy, producer accuracy, and Kappa coefficient. The classifier, carried out using 58, 90, and 211 points, was generated randomly for 1998, 2008, and 2018 supervised images, respectively. Each point has a specific pixel value based on ground-truth data and visual interpretation. Identified points were considered classified values, and pixel values were considered reference values.

Classification results were carried out on thematic maps using different truth reference data, and accuracy assessments were presented for all classifications. Overall accuracy, user accuracy, producer's accuracy, and Kappa coefficient accuracy derived from the error matrices were used for the accuracy assessment of the final maps. An error matrix and Kappa statistics were generated from reference and classified data.

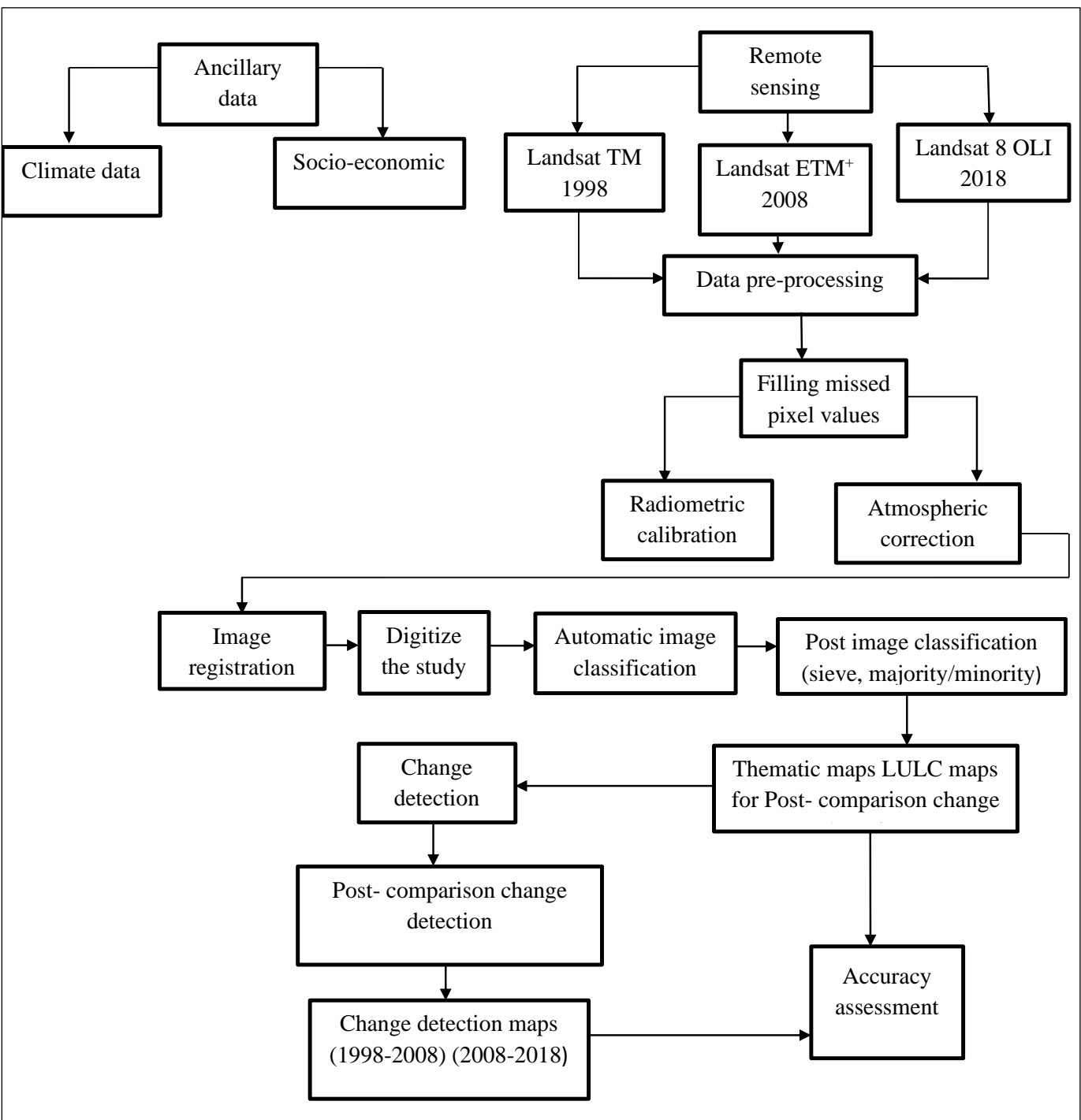

**Figure 2.** Flow chart of the remote sensing methodology.

In this study, satellite images were accurately assessed with the Landsat images Landsat 5-TM 1988, Landsat 7-ETM+ 2008, and Landsat 8 OLI 2018, which the ground truth data likely equates to. Overall accuracy was calculated by dividing the sum of correctly classified pixels (diagonal) [39] by the total number of reference pixels.

$$\text{Overall accuracy} = (\textstyle\sum\nolimits_{(e\,=\,1)}^{L} ncc)/N \tag{1}$$

The overall classification accuracy was calculated for all classifications, as well as class-specific accuracy, which can be created on the user and producer levels. Users' accuracy, i.e., the error of commission [40] and user accuracy assessment, was calculated by dividing

the number of correctly classified pixels in each category by the total number of classified pixels in that category (row total).

$$\text{Users' accuracy} = ncc/(n + c) \tag{2}$$

Producer accuracy is the measure of the error of omission [40], and producer accuracy was calculated by dividing the number of correctly classified pixels in each category by the total number of reference pixels in that category (total column).

$$\text{Producer accuracy} = ncc/(nc+) \tag{3}$$

The Kappa coefficient was calculated by using the following equations:

$$k\hat{} = (\sum\nolimits_{(1\text{-}1)}^r \llbracket P\_ii -\sum\nolimits_{(1\text{-}1)}^r \llbracket P\_(i+) \times P\_(+i) \rrbracket \rrbracket )/(1\text{-}\sum\nolimits_{(1\text{-}1)}^r P\_(i+) \times P\_(+i)) \tag{4}$$

where r = the number of rows in the error matrix;

P_ii = the proportion of pixels in row 'r' and column 'i';
P_(i+) = the proportion of the marginal total of row 'i';
P_(+i) = the proportion of the marginal total of column 'i'.

### 2.6. Cellular Automata Model

The cellular automata model mainly consists of cell, time, and rule neighbor. The filter of the cellular model determines the neighbors. Cellular automata (CA) are a spatiotemporal calculation of a dynamic process model used for LULC change. The commonly used neighborhoods are Moore, the extended Moore, and von Neumann. The expression of the Cellular Automata equation is:

$$S (t, t + 1) = f (St, N) \tag{5}$$

where S is the set of states of the finite cells, the (t, t + 1) are different times, N is the neighborhood of cells, and f is the transformation rule of local space [41].

### 2.7. CA–Markov Model

The CA–Markov model is valuable for modeling LULC changes and can simulate prediction changes. The spatial prediction accuracy can be effectively simulated simultaneously [42]. CA–Markov model effectively combines the Markov model and the CA model. This approach is based on a Markov stochastic probability matrix for predicting the transition from one condition to another [30]. The Markov chain model is most commonly used to simulate transitions, parameters, and trends. It created probability transition matrices to anticipate and categorize probable land use/cover change (LUCC) and urban development scenarios and investigate land change simulation trends [30]. Equations ((6)–(8)) based on the conditional probability formula were used to estimate trends:

$$(t + 1) = P_{ij} \times S(T), \tag{6}$$

$$P_{ij} = P_{ij} = \begin{pmatrix} P_{11} & P_{12} & P_{1n} \\ P_{21} & P_{22} & P_{2n} \\ P_{n1} & P_{n2} & P_{n3} \end{pmatrix}, \tag{7}$$

$$(0 \leq P_{ij} < 1 \; and \; \sum\nolimits_{j=1}^{N} P_{ij} = 1, \; (i, j = 1, 2, \ldots \ldots n), \tag{8}$$

where S(t) is the state of the system at time t, S (t + 1) is the state of the system at time (t + 1), and Pij is the matrix of the transition probability in a state. By projecting 2018 and 2048, the cellular automata (CA) and Markov chain models are employed to generate the LUCC

future scenario. The land use change modeler (LCM) in TerrSet (Clark Labs TerrSet 18.31) was used to forecast LUCC for the projected time using the CA–Markov model [30].

### 2.8. Landscape Metrics

The FRAGSTATES software version 4.2.1 [43] was applied for computed landscape metrics in this study area. Landscape metrics characterized the spatial fragmentation and heterogeneity for 1998, 2008, and 2018, covering three decades. The selected landscape patterns were divided into three levels, i.e., patch, class, and landscape. FRAGSTATS is a spatial pattern analysis program implemented by the decision maker, forest manager, and ecologists to analyze landscape fragmentation or describe characteristics of the landscape and elements of those landscapes [44]. FRAGSTATS version 4.2.1 was used to extract the landscape metrics from each 1998, 2008, and 2018 classified map. In this study, a total of twelve landscape metrics were examined, including:

- Class level: Class Area (CA), Total Landscape Area (TLA), Number of Patches (NP), Mean Patch Size (MPS), Patch Size Coefficients of Varian (PSCOV), Mean Shape Index (MSI), Edge Density (ED), Mean Nearest Neighbor Distance (MNN), and Interspersion Juxtaposition Index (IJI);
- Landscape level: Total Landscape Area (TLA), Number of Patches (NP), Mean Patch Size (MPS), Mean Shape Index (MSI), Mean Nearest Neighbor Distance (MNN), Interspersion Juxtaposition Index (IJI), Mean Proximity Index (MPI), Shannon's Diversity Index (SDI), and Shannon's Evenness Index (SHEI).

### 2.9. LULC Change-Driving Forces Model

Previous studies have suggested that the main LULC of the Chittagong Hill Tracts area driving factor is climate change, while anthropogenic contributed to LULC slightly. Therefore, this study only considered climate change as a leading driving factor.

The monthly meteorological data from 1998 to 2018 were collected from the meteorological station of Bangladesh. Each year of the CHT area was obtained by substituting time into the linear regression model. The annual average temperature (AAT), annual average relative humidity (AARH), and annual average rainfall (AAR) were used in the analysis. Climate change factors such as the average yearly temperature and annual average relative humidity were used to establish corresponding data sets. All the meteorological data processing steps were carried out in linear regression.

Climate change factors such as the annual average temperature and annual average relative humidity were used to establish corresponding data sets. The relationship between dependent variables and independent variables is given below:

$$Y = a + b_1 x_1 + b_2 x_2 + b_3 x_3 + \ldots \ldots \ldots \ldots + b_n x_n, \tag{9}$$

where Y is a dependent variable, a is a constant turn, $x_1$, $x_2$, and $x_3$ are independent variables, and $b_1$, $b_2$, and $b_3$ are the coefficients of independent variables.

The constant coefficient illustrates the value of Y in all dependent variables at a zero time zone. In addition, the parameter coefficients state a change in Y for one unit increase in the dependent variable [45]. T-tests were applied to verify the statistical regression coefficients' significance and constant in the linear regression model. The determination sample coefficient ($R^2$) was applied to explain the contribution to the dependent variable in the linear regression model. $R^2$ means how closely dependent variables are related to independent variables [46].

### 3. Results

#### 3.1. LULC Classes and Their Distribution

Land use cover changes (LULCCs) were computed for 1998, 2008, and 2018, focusing on vegetation, grassland, water bodies, and the bare land area in the study area (Figure 3). In 1998, the maximum land area covered by grassland accounted for 44.71% (5901.11 Km$^2$),

which gradually decreased to 43.90% (5794.51 Km$^2$) in 2008 and 24.10% (3180.34 Km$^2$) in 2018. Whereas the forest area was calculated to be approximately (5576.26 Km$^2$) in 1998, 5048.22 Km$^2$ (2008), and 8284.62 Km$^2$ (2018), with a slight decrease of $-4\%$ from 1998 to 2008 and periodic increments of 24.52% from 2008 to 2018. The bare land area was calculated to be 1188.75 Km$^2$ in 1998, 1853.79 Km$^2$ in 2008, and 1161.72 Km$^2$ in 2018, with an increase of 5.04% from 1998 to 2008 and then a decrease of $-5.25\%$ from 2008 to 2018 (Table 3). The cumulative change calculated in water area was approximately 39.44 Km$^2$ (1998–2018), 532.04 Km$^2$ in 1998, and 571.48 Km$^2$ in 2018. An increase of 0.3% was observed in water from 1998 to 2018 (Table 3, Figure 4).

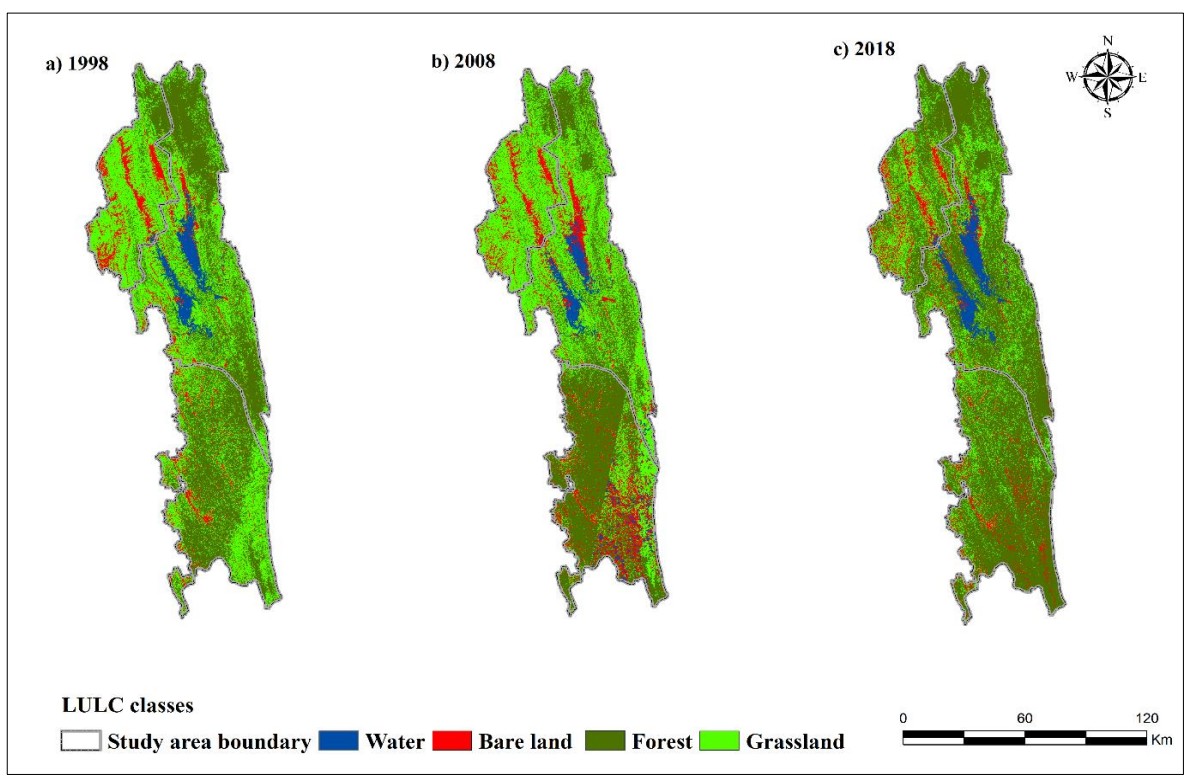

**Figure 3.** LULC thematic maps of Chittagong Hill Tracts: (**a**) 1998, (**b**) 2008, and (**c**) 2018.

**Table 3.** Depicting the percentage of the spatial coverage of LULC in Chittagong Hill Tracts from 1998 to 2018.

| Year | Land Use Classes | | | | | | | | | |
|------|-------|---|--------|---|-----------|---|----------|---|-------|---|
| | **Water** | | **Forest** | | **Grassland** | | **Bare land** | | **Total** | |
| | Area | | | | | | | | | |
| | **Km$^2$** | **%** | **Km$^2$** | **%** | **Km$^2$** | **%** | **Km$^2$** | **%** | **Km$^2$** | **%** |
| 1998 | 532.04 | 4.03 | 5576.26 | 42.25 | 5901.11 | 44.71 | 1188.75 | 9.01 | 13,198.16 | 100 |
| 2008 | 501.64 | 3.80 | 5048.22 | 38.25 | 5794.51 | 43.90 | 1853.79 | 14.05 | 13,198.16 | 100 |
| 2018 | 571.48 | 4.33 | 8284.62 | 62.77 | 3180.34 | 24.10 | 1161.72 | 8.80 | 13,198.16 | 100 |

Kappa coefficients and user accuracy for supervised classification (LULCC maps) were calculated with TerrSet IDRISI. The overall accuracy of the classification was observed at 89.65%, 84.44%, and 86.26%, while producer accuracies were 90.00%, 68.75%, and 72.22%, and the Kappa coefficient was 85.68, 82.84, and 76.36 in three different periods (Table 4), respectively.

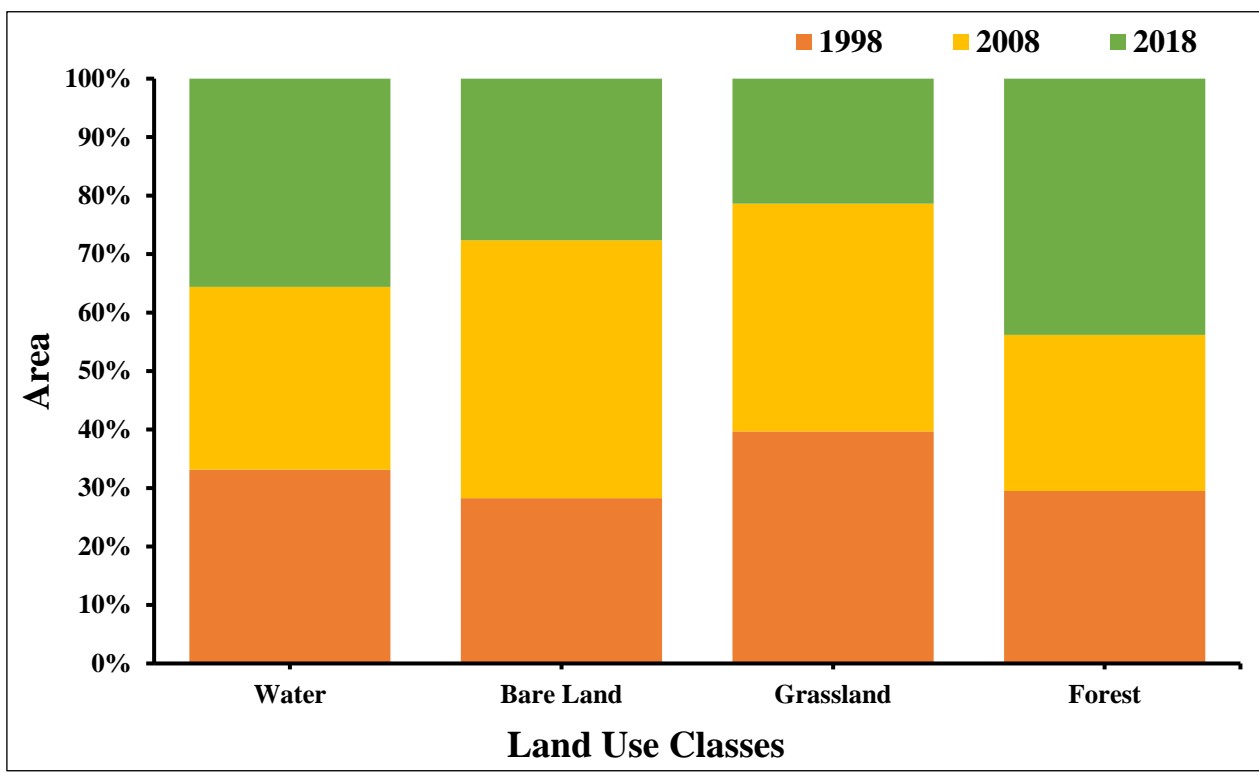

**Figure 4.** Spatial area coverage % of LULC in Chittagong Hill Tracts during the study interval (1998–2018).

**Table 4.** Accuracy computation of land use and land cover change (LULC) maps between 1990 and 2018.

| Year | LULC Classes | Water | Bare Land | Forest | Grassland | Kappa Value | Overall Accuracy |
|------|-------------|-------|-----------|--------|-----------|-------------|------------------|
| 1998 | User's | 81.82 | 77.78 | 95.00 | 94.44 | 0.857 | 89.65 |
|      | Producer's | 90.00 | 77.78 | 95.00 | 89.47 | | |
| 2008 | User's | 84.61 | 64.28 | 91.17 | 86.21 | 0.828 | 84.44 |
|      | Producers | 68.75 | 69.23 | 91.18 | 92.59 | | |
| 2018 | User's | 86.67 | 76.19 | 92.06 | 75.51 | 0.764 | 86.26 |
|      | Producer's | 72.22 | 66.66 | 92.10 | 86.04 | | |

*3.2. LULC Change Detection*

The post-classification comparison of LULC change for each class within the study period from 1998 to 2018 is shown in Figure 5, whereas the result analysis of LULC change is indicated in Table 5. From 1998 to 2008, only bare land increased by 665.04 Km². In contrast, water, forest, and grassland decreased significantly by −30.40 Km², −528.04 Km², and −106.60 Km², respectively, with a significant decrease in bare land and grassland, i.e., −692.07 Km² and −2614.17 Km², between 2008 and 2018 (Table 5). During the three decades, the grassland decreased by −2720.77 Km², while the forest increased by an area of about 2708.36 Km².

Regarding LULC conversions and transformations, the most dynamic change in the study area is the conversion of grassland to forest by 13.34% (Table 6), followed by forest into grassland by nearly 58.03%. The transformation of bare land to forest was 1.68%, and the transformation of bare land into grassland was calculated to be 6.28%. However, forest land conversion to bare land in the region was estimated to be 12.10%, while the transformation of grassland into bare land was observed to be 29.50%. Therefore, between 1998 and 2018, the region experienced many changes in its LULC.

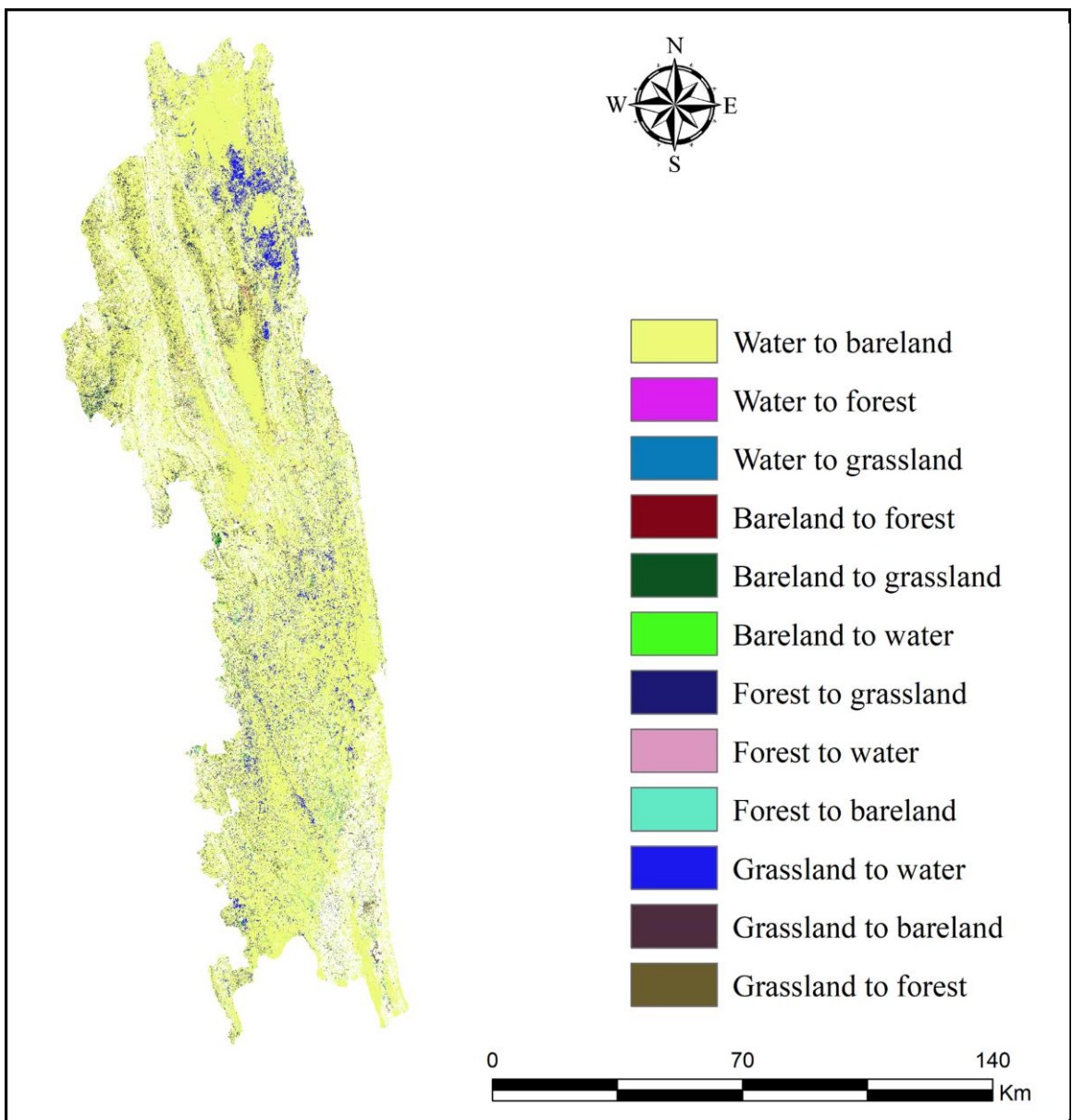

**Figure 5.** LULC changes in Chittagong Hill Tracts from 1998 to 2018.

**Table 5.** Analysis of LULC change in Chittagong Hill Tracts.

| Classes | Net of LULC Changed (Km²) | | |
|---|---|---|---|
| | **1998–2008** | **2008–2018** | **1998–2018** |
| Water | −30.40 | 69.84 | 39.44 |
| Bare land | 665.04 | −692.07 | −27.03 |
| Forest | −528.04 | 3236.40 | 2708.36 |
| Grassland | −106.60 | −2614.17 | −2720.77 |

**Table 6.** Change matrices of LULC classes in Chittagong Hill Tracts between 1998 and 2018.

| Year | | 1998 | | | |
|---|---|---|---|---|---|
| | | **LULC Classes** | | | |
| | | **Water** | **Bare land** | **Forest** | **Grassland** |
| | Area | % | % | % | % |
| **2018** | **Water** | 95.72 | 1.26 | 0.73 | 0.51 |
| | **Bare land** | 2.73 | 57.59 | 1.68 | 6.28 |
| | **Forest** | 1.36 | 12.10 | 84.25 | 58.03 |
| | **Grassland** | 0.19 | 29.50 | 13.34 | 35.19 |

*3.3. Prediction of LULC Change Based on the Markov Model*

The state transition area map was created according to LULC maps from 1998 to 2018, which can be used to predict, using the CA–Markov model in IDIRISI software version 17.0, the land requirements for the different LULC types in 2048. The predictive results map for 2018 is obtained with a $5 \times 5$ contiguity filter, whose running cycle is 30 years. The combination of cellular automata (CA) and the stochastic transition matrix of the Markov chain model resulted in LUCC for the projected period of 2048 (Figure 6). Map accuracy for the projected land use/cover change for predictive years was classified by a sufficient Kappa coefficient value of 0.97. A 2048 map predicted that the maximum area covered by forest accounts for 9129 Km$^2$ (69.17%) and the minimum area covered by water accounts for 665 Km$^2$ (5.05%). A decrease of 1.51% of bare land and 5.61% of grassland cover areas were estimated during 2018–2048, respectively (Table 7). However, forest and water area cover will expand by 6.40% and 0.72%, respectively, during 2018–2048.

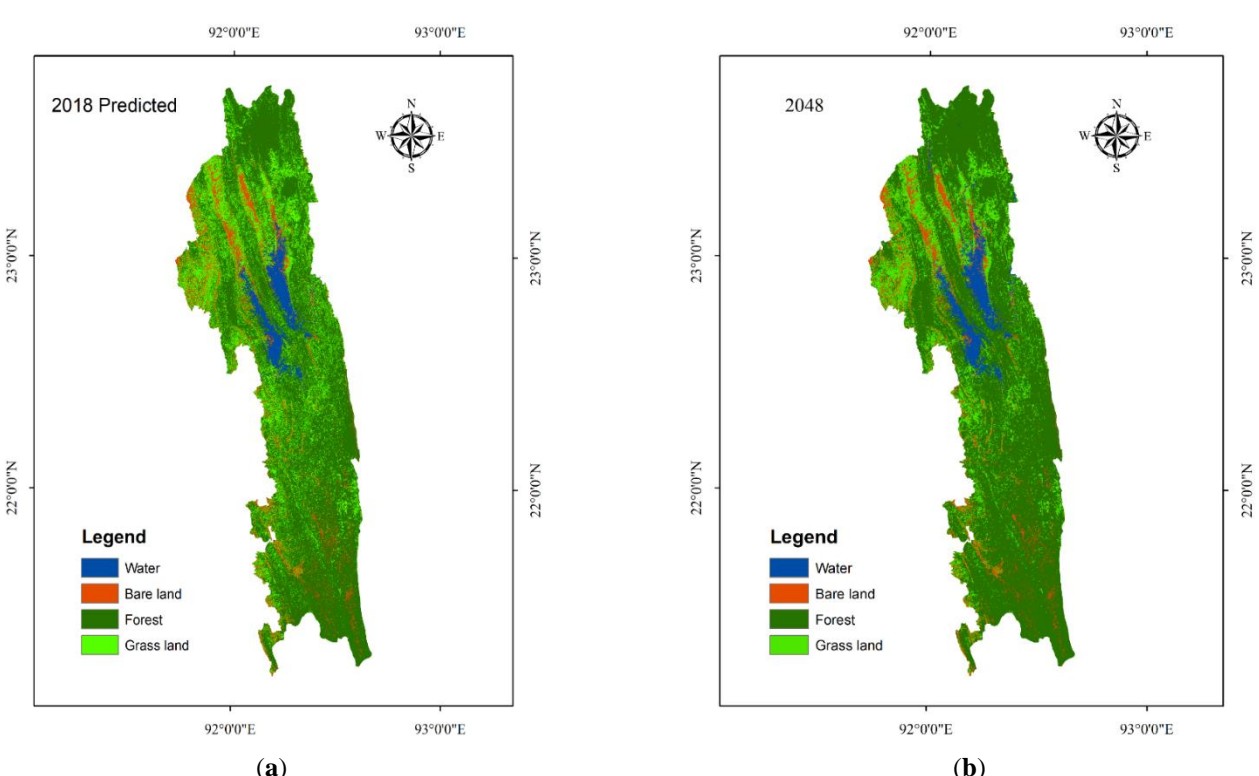

**Figure 6.** Predicted LULC classification maps of Chittagong Hill Tracts (**a**) predicted map of 2018 and (**b**) predicted map of 2048.

**Table 7.** Probability area percentage of LULC classes in Chittagong Hill Tracts in 2048.

| | Period | | | |
|---|---|---|---|---|
| **LULC Classes** | **2048** | | **Change Detection 2018–2048** | |
| | **Km²** | **%** | **Km²** | **%** |
| Water | 665 | 5.05 | 93.52 | 0.72 |
| Bare land | 963 | 7.29 | −198.72 | −1.51 |
| Forest | 9129 | 69.17 | 844.38 | 6.40 |
| Grassland | 2441 | 18.49 | −739.34 | −5.61 |
| Total area | 13,198 | 100 | | |

*3.4. Landscape Metrics Analysis of Land Use and Land Cover Structure*

From LULC maps from 1998 to 2018, the most changing classes, such as water, bare land, forest, and grassland, were chosen to analyze spatial landscape metrics at class and landscape levels. The most significant change among land use and land cover classes is increased forest and decreased grassland. The statistical results of the landscape metrics in the Chittagong Hill Tracts area are shown below (Table 8).

**Table 8.** Metrics of landscape patterns for 1998, 2008, and 2018.

| LULC Classes | Year | Indices | | | | | |
|---|---|---|---|---|---|---|---|
| | | **CA (ha)** | **NP** | **ED (m/ha)** | **IJI (%)** | **PSCOV (ha)** | **MPS (ha)** |
| Water | 1998 | 58,223.88 | 217 | 1.2 | 92.32 | 831.16 | 235.37 |
| | 2008 | 58,223.88 | 739 | 2.16 | 93.48 | 991.11 | 78.79 |
| | 2018 | 57,432.96 | 256 | 1.33 | 95.07 | 925.63 | 224.35 |
| Bare land | 1998 | 119,352.9 | 2030 | 5.5 | 60.38 | 669.55 | 58.79 |
| | 2008 | 178,306.8 | 2671 | 8.56 | 85.47 | 665.78 | 66.76 |
| | 2018 | 116,615.1 | 2431 | 6.3 | 79.38 | 473.06 | 47.97 |
| Forest | 1998 | 557,963.6 | 1639 | 13.7 | 36.37 | 2388.84 | 340.43 |
| | 2008 | 505,352.3 | 2049 | 13.2 | 57.75 | 2969.51 | 246.63 |
| | 2018 | 828,093.2 | 744 | 16.09 | 59.96 | 2548.7 | 1113.03 |
| Grassland | 1998 | 591,653.8 | 1704 | 17.07 | 63.85 | 2723.98 | 347.21 |
| | 2008 | 578,162.5 | 1492 | 15.7 | 73.15 | 3197.04 | 387.51 |
| | 2018 | 317,904.2 | 3268 | 15.17 | 52.48 | 1057.64 | 97.28 |

CA = Class Area (ha); NP = Number of Patches; ED = Edge Density (m/ha); IJI = Interspersion Juxtaposition Index (%); PSCV = Patch Size Coefficient of Varian (ha); MPS = Mean Patch Size (ha).

The statistics of water showed that Class Area (CA) indices increased from 51,075.18 to 57,432.96 ha, the Number of Patches (NP) also increased from 217 to 256, the Patch Size Coefficient of Varian (PSCV) increased from 831.16 to 925.63 ha, and Mean Patch Size (MPS) decreased from 235.37 to 224.35 ha during the whole period from 1998 to 2018.

Regarding bare land area during the period 1998 to 2018, CA indices decreased from 119,352.9 to 116,615.1 ha, NP increased from 2030 to 2431, Edge Density (ED) increased from 5.5 to 6.3 m/ha, Interspersion Juxtaposition Index (IJI) increased from 60.38 to 79.38%, while the PSCV decreased from 669.55 to 473.06 ha, and MPS decreased from 58.79 to 47.97 ha.

The spatial analysis of forest areas showed that CA indices increased from 557,963.6 to 828,093.2 ha, NP decreased from 1639 to 744, ED increased from 13.7 to 16.09 m/ha, IJI increased from 36.37 to 59.96%, PSCOV increased from 2388.84 to 2548.7 ha, and MPS decreased from 58.79 to 47.97 ha.

Grassland areas showed that CA indices decreased from 591,653.8 to 317,904.2 ha, NP increased from 1704 to 3268, ED decreased from 17.07 to 15.17 m/ha, IJI decreased from

63.85 to 52.48%, PSCOV decreased from 2723.98 to 1057.64 ha, and MPS decreased hugely from 347.21 to 97.28 ha.

The analysis of the landscape level showed that the fragmentation of the landscape increased with the number of patches (NP) from 5590 to 6699, the Mean Proximity Index (MPI) increased from 2296.4 to 2676.39 m/ha, the Mean Patch Size (MPS) decreased from 236.14 to 197.05, Mean Shape Index (MSI) values are identical, Mean Nearest Neighbor Distance (MNND) decreased from 606.1 to 567.5 m, Interspersion Juxtaposition Index (IJI) increased from 56.95 to 62.83%, Shannon's Diversity Index (SEI) decreased from 0.77 to 0.71, and Shannon's Evenness Index (SDI) decreased from 1.07 to 0.99 (Table 9). There is no significant change in the heterogeneity of the landscape.

**Table 9.** Analysis matrices of landscape patterns with references to the studied years 1998, 2008, and 2018.

| Year | Indices | | | | | | | |
|---|---|---|---|---|---|---|---|---|
| | NP | MPI | MPS | MSI | MNND | IJI | SEI | SDI |
| **1998** | 5590 | 2296.4 | 236.14 | 1.27 | 606.1 | 56.95 | 0.77 | 1.07 |
| **2008** | 6951 | 2231.18 | 189.91 | 1.23 | 583.3 | 72.68 | 0.82 | 1.14 |
| **2018** | 6699 | 2676.39 | 197.05 | 1.27 | 567.5 | 62.83 | 0.71 | 0.99 |

NP = Number of Patches; MPI = Mean Proximity Index; MPS = Mean Patch Size (ha); MSI = Mean Shape Index; MNND = Mean Nearest Neighbor Distance; IJI = Interspersion Juxtaposition Index (%); SEI = Shannon's Diversity Index; SDI = Shannon's Evenness Index.

### 3.5. Analysis of the Driving Forces behind LULC Change

The linear regression model result indicated that all driving factors had significant values in adjusted R square for the model of change in the forest, bare land, grassland, and water of 0.619, 0.559, 0.718, and 0.752, respectively (Table 10). The study's results explained that AT was highly significant and AR substantially impacted forests ($p < 0.01$, $p < 0.05$; Tables 8 and 11). AR was highly influential, and ARH significantly impacted bare land ($p < 0.01$, $p < 0.05$; Tables 8 and 11). AT had a highly significant impact on grassland ($p < 0.01$; Tables 8 and 11). AT had a highly substantial effect, and ARH, AR had a considerable effect on water ($p < 0.01$, $p < 0.05$; Tables 8 and 11) and the change of CHT in Bangladesh.

**Table 10.** Climate change impact driving model variables included in the CHT.

| Model | $R^2$ | Std Error | F Value | Sig. |
|---|---|---|---|---|
| Y1 | 0.62 | 611.37 | 9.12 | 0.008 ** |
| Y2 | 0.56 | 138.88 | 7.35 | 0.015 * |
| Y3 | 0.72 | 462.73 | 13.75 | 0.002 ** |
| Y4 | 0.75 | 10.49 | 11.10 | 0.004 ** |

Note: Y1 = forest land, Y2 = bare land, Y3 = grassland, and Y4 = water. ** = highly significant; * = significant.

**Table 11.** Estimating the influence of drivers behind changes in LULC.

| Model | | Std Error | Coefficients | T Value | Sig. |
|---|---|---|---|---|---|
| | Intercept | 3883.059 | −6113.87 | −1.5745 | 0.154 [NA] |
| Y1 | AR | 0.644799 | −1.51007 | −2.34193 | 0.047 * |
| | AT | 159.0128 | 658.4076 | 4.140596 | 0.003 ** |
| | Intercept | 1046.292 | −2152.03 | −2.05681 | 0.073 [NA] |
| Y2 | AR | 0.144635 | 0.510366 | 3.528659 | 0.007 ** |
| | ARH | 13.31509 | 32.69115 | 2.455195 | 0.039 * |
| | Intercept | 2938.974 | 18,078.93 | 6.151443 | 0.000 ** |
| Y3 | AR | 0.488029 | 1.026891 | 2.104158 | 0.068 [NA] |
| | AT | 120.3521 | −629.188 | −5.22789 | 0.000 ** |

**Table 11.** *Cont.*

| Model | | Std Error | Coefficients | T Value | Sig. |
|---|---|---|---|---|---|
| | Intercept | 156.4302 | −159.272 | −1.01817 | 0.342 [NA] |
| Y4 | AR | 0.011129 | 0.026354 | 2.367997 | 0.049 * |
| | ARH | 1.308958 | 3.491847 | 2.667654 | 0.032 * |
| | AT | 3.551116 | 16.20295 | 4.562777 | 0.002 ** |

Note: Y1 = forest land, Y2 = bare land, Y3 = grassland, and Y4 = water. AR = average rainfall, AT = average temperature, and ARH = average relative humidity. ** = highly significant; * = significant; [NA] = non-significant.

## 4. Discussion

### 4.1. Accuracy Assessment

After calibration, land cover data were extracted from photographs and corrected for atmospheric effects. Landsat 5 TM 1998 had six spectral bands plotted upon them, but Landsat 7 ETM+ 2008 and Landsat 8 OLI 2018 had seven bands, respectively. In three different periods, a summary of supervised classification accuracy was recorded as the highest in 1998 (89.65%) and 2008 (84.44%). This result is consistent with the study findings by the authors [44]. The study indicated that all classes of producers were classified over 85% accurately. Similarly, a study by Dewan and Yamaguchi [47] using MSS found that the lowest overall accuracy was 85.6%. Another study by Kayiranga et al. [48] revealed that from 1986 to 2015, all images' overall accuracy and Kappa coefficient results were greater than 75%. In this study, among the three periods of 1998, 2008, and 2018, Kappa coefficients were above 0.8% in all classes except 2018. Very similar results were found in the study conducted by del Castillo [49], who investigated that all forest classes' Kappa coefficients were above 0.8 except *Q. pyrenaica* in 2010.

### 4.2. Trends in Forest Cover Changes

According to estimates of forest cover change (loss and gain) derived from Landsat images during the past twenty years, forest and tree cover increased from 1998 to 2018. The national assessment led by FAO [50] showed that the net growth of tree cover has remained stable since 1990. Different spatial and temporal extents may, in part, explain these discrepancies. Furthermore, while our results show a net gain in forest cover over time, details on the visibility and psychological impact [51] on forest loss vs. gain may also shape perspectives on forest dynamics [52].

The most notable LULC changes were the transformation of grassland to forest. The analysis of the Landsat images reveals some trends that complement the LULC from 1998 to 2018. From 1998 to 2018, the change in LULC classes shows that the Chittagong Hill Tracts region has been impacted and degraded with forest (2708.36 $Km^2$) and bare land (−27.03 $Km^2$), respectively. A similar result was contrary to the range reported elsewhere [53,54]. A similar result was proposed by Baral et al. [55], who reported that forest cover increased with a similar loss in the extent of all other land cover types. The human population was the most important factor in increasing or decreasing land use and land cover change.

Here is the evidence that variations in forest cover changes may be associated with the types of management regimes [56]. Community-based management or secure tenure can result in positive social interactions, better forest conditions, and opportunities to increase tree cover and distribute economic benefits [57]. State-led plantation initiatives and the National Forest Policy of 1994 have been primarily responsible for expanding forests in the CHT region, focusing on plantations made possible by the relocation of native peoples [58]. Bangladesh has demonstrated encouraging progress in curbing the annual deforestation rate. It has reduced deforestation from 2.1% from 1960 to 1980 [59] to approximately 0.2% between 1990 and 2010 [18].

### 4.3. Future LULC Change Prediction

In contrast, between 2018 and 2048, the predicted LULC in Chittagong Hill Tracts grassland significantly decreased after 1998, the forest increased, and the remaining area

increased by a certain amount in 2048. Similar predictive results were proposed by Liping et al. [41]; they reported that water and bare land increased and the woodland area decreased significantly. Predictive results were revealed by [60]: bare land will increase and potentially lead to the loss of ecosystem services. The Markov predictive results have shown that the probability of each class changing in LULC in the future is high.

### 4.4. Landscape Analysis

During the study period (1998–2018) in Chittagong Hill Tracts, changes occurred in four major land classes: water, bare land, forest, and grassland. The results of the landscape indices (forest) revealed that the values of NP decreased proportionally with increased CA, ED, IJI, PSCOV, and MPS. Similar results were found in the study conducted by Kayiranga et al. [46], who reported that during the whole study period (1986–2015), the values of MPS increased with a decrease of NP inside the Nyungwe–Kibira Park. del Castillo et al. [49] also reported that the forest inside the Moncayo Natural Park (Spain) was slightly fragmented. This study claims landscape pattern metrics at heterogeneity with a similar identic landscape configuration. According to research correlations on landscape metrics, Shannon's Evenness Index (SEI) and Shannon's Diversity Index (SDI) were also selected as indices for the landscape pattern analysis [61]. During the study period, Shannon's Diversity Index and Shannon's Evenness Index showed a slightly decreased landscape. The landscape metrics analysis provided valuable information about land use and land cover change, especially fragmentation and heterogeneity weighed at the landscape class and landscape level. The landscape structure became more fragmented and heterogeneous [62]. Moreover, during the study period, landscape fragmentation increased and forests became a major land area [63,64]. Landscape (fragmentation and heterogeneity) and land use/landcover results indicated that forest management significantly increases, decreases, and maintains landscape patterns [49].

### 4.5. LULC Change–Climate Factors

The linear regression results assumed that the climate change factors model variables indicated that three impact factors were selected for the forest: bare land, grassland, and water. These three factors are less than significantly related to forest, bare land, grassland, and water. According to Salman-Mahini and Kamyab [65], there was a linear relationship between dependent and independent variables in multiple linear regression. The result of linear regression indicated that the climate change factors (average rainfall, average humidity, and average temperature) model as independent variables estimated the influence of drivers behind changes in four kinds of LULC change: forest, bare land, grassland, and water. Grassland was most affected by rainfall, the forest was significantly affected by rainfall and temperature, bare land was significantly affected by rainfall and humidity, and water was significantly affected by rainfall, humidity, and temperature. Rainfall was the main factor driving LULC changes. A similar result was found in a Northern Iranian study by Jahanifar et al. [46]. The linear alternative of income per capita, rain, and temperature with a definite coefficient of 0.4 as independent variables qualified for estimating forest area reduction.

Although research into the spatial-temporal trends and connections between forest cover and local land use practices will aid in identifying regions where changes have taken place and making predictions about future changes, this research alone will not be sufficient to explain the neighborhood's driving factors of change, which are necessary for making trained, evidence-based decisions. Understanding the mechanisms of forest cover changes, like deforestation and forest regeneration, in the Chittagong Hill Tracts at various geographical scales requires knowledge of the local driving variables at work. Therefore, future research should investigate the causes and consequences of land use and forest cover changes within this landscape's socioeconomic, policy, institutional, and past histories and how these shifts relate to larger shifts happening at the regional and global levels.

## 5. Conclusions

Knowledge of past land use trends and land cover change is vital to comprehending the connection between landscape dynamics and ecological responses. Our method, which integrates multi-temporal remote sensing data with GIS techniques, has enabled us to quantify and characterize the spatial-temporal pattern of LULC changes, notably those associated with forest cover changes in mountainous regions. The findings revealed that the terrain of the Chittagong Hill Tracts underwent LULC shifts between 1998 and 2018. Between those years, forest cover and water body area expanded while grassland and bare land shrank. The spatial pattern shift demonstrated that between 1998 and 2008 and between 2008 and 2018, the forest cover gained and lost area with varying annual intensities and dynamics.

The annual increase was significant during both periods, while deforestation was prominent during the first period (1998–2008) but inactive during the second period (2008–2018). Between 1998 and 2018, there was a net change of 2708.36 $Km^2$ in forest cover and a total change of 8284.62 $Km^2$. The LULC structure, predicted based on the CA–Markov model that in 2048, the forest area would increase drastically with a consistent decrease in grassland and bare land areas. The linear regression results for the climate change factors model variables indicated that grassland was most affected by rainfall, the forest was significantly affected by rainfall and temperature, bare land was significantly affected by rainfall and humidity, and water was significantly affected by rainfall, humidity, and temperature. Rainfall was the main factor driving LULC changes. Currently, the area faces different environmental issues that threaten these resources, such as climate change, LULC change, disturbance of species diversity, ecosystem fragmentation, and flooding. It was found that massive population growth in settlements is an important factor influencing LULC changes and the implications of sustainable landscape management. Furthermore, the findings of this study have some important suggestions that a scientific forest management system should have a strict process. Additionally, efficient programs should be run to educate local communities about sustainable land management.

**Author Contributions:** Conceptualization, M.C., U.H. and J.M.; methodology, M.C. and U.H.; software, M.C., U.H. and M.A.H.; validation, M.C., U.H. and J.M.; formal analysis, M.C. and U.H.; investigation, M.C.; resources, U.H. and J.M.; data curation, M.A.H.; writing—original draft preparation, M.C. and U.H.; writing—review and editing, U.H. and J.M.; visualization, U.H.; supervision, J.M.; project administration, J.M.; funding acquisition, J.M. All authors have read and agreed to the published version of the manuscript.

**Funding:** This research was funded by the National Natural Science Foundation of China (32271871).

**Data Availability Statement:** Data will be available on demand.

**Acknowledgments:** We would like to thank all the lab members of the Research Center of Forest Management Engineering of the National Grassland Administration, Beijing Forestry University, for their technical and moral support.

**Conflicts of Interest:** The authors declare no conflict of interest.

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
