# Peer review of "An Assessment of Landscape and Land Use/Cover Change and Its Implications for Sustainable Landscape Management in the Chittagong Hill Tracts, Bangladesh"

_land, doi:10.3390/land12081610_

Round 1

Reviewer 1 Report

Line 30. "... slight decrease of the landscape." Maybe these are translation errors. Revise.

Line 32. ".. other land uses continue to decrease". Maybe need some clarification such us land use types/classes or categories as mentioned by authors. 

Line 53. It is necessary to rephrase the sentence as it is not clear 'continuous changes and dynamics...". 

Study area. Geographical coordinates of the research area have to be corrected and provided in correct form. 

Fig. 1. Does the provided scale correspond to both images?

Line 175. "In this study, level-1 downloaded images were rectified and corrected", better to be corrected. Landsat Collections Level-1 images ...

Line 210. Why accuracy assessment done only with OLI 2018? Maybe give an explanation.

Line 254-255. "The selected landscape patterns were 254 divided into three levels, i.e., the patch, class, and landscape." One may consider employing an alternative definition as the stylistic use of two identical descriptions could potentially perplex the reader.

Table 11. Capitalise "estimating"

Slightly changes and corrections are needed. Some correction recommendations as an example are provided in the comment section. 

Author Response

Reviewer-1

Line 30. "... slight decrease of the landscape." Maybe these are translation errors. Revise.

Ans: Revised.

Line 32. ".. other land uses continue to decrease". Maybe need some clarification such us land use types/classes or categories as mentioned by authors.

Ans: Revised.

Line 53. It is necessary to rephrase the sentence as it is not clear 'continuous changes and dynamics...".

Ans: Revised.

Study area. Geographical coordinates of the research area have to be corrected and provided in correct form.

Ans: Figure.1 has been revised.

Fig. 1. Does the provided scale correspond to both images?

Ans: Figure.1 has been revised.

Line 175. "In this study, level-1 downloaded images were rectified and corrected", better to be corrected. Landsat Collections Level-1 images ...

Ans: Modified as suggested.

Line 210. Why accuracy assessment done only with OLI 2018? Maybe give an explanation.

Ans: It was a typo, actually in this study, an accuracy assessment of satellite images was done with the Landsat TM 1988, ETM+ 2008 and Landsat 8 OLI 2018.

Line 254-255. "The selected landscape patterns were 254 divided into three levels, i.e., the patch, class, and landscape." One may consider employing an alternative definition as the stylistic use of two identical descriptions could potentially perplex the reader.

Ans: In this study area, the FRAGSTATES was applied for computed landscape metrics. Landscape metrics were used to characterize the spatial fragmentation and heterogeneity for 1998, 2008 and 2018 covering three decades. FRAGSTATS is a spatial pattern analysis program was implemented by decision maker, forest manager and ecologists to analyze landscape fragmentation or describing characteristics of landscape, elements of those landscapes (KeleÅŸ et al., 2008). The software FRAGSTATS version 4.2.1 was used to extract the landscape metrics from each classified map of 1998, 2008 and 2018. In this study, total twelve landscape metrics including:

- class level: class area (CA), total landscape area (TLA), number of patch (NP), mean patch size (MPS), patch size coefficients of Varian (PSCOV), mean shape index (MSI), edge density (ED), mean nearest neighbor distan (MNN), Interspersion Juxtaposition Index (IJI),

- landscape level:total landscape area (TLA), number of patch (NP), mean patch size (MPS), mean shape index (MSI), mean nearest neighbor distan (MNN), Interspersion Juxtaposition Index (IJI), Mean Proximity Index (MPI), Shannon’s Diversity Index(SDI), Shannon’s Evenness Index (SHEI).

Table 11. Capitalise "estimating"

Ans: Corrected.

Reviewer 2 Report

(1) It is suggested that in the introduction section, the limitations of the current research as well as the innovations of this study should be added.

(2) The authors only introduce the remote sensing data, and lack of explanation on the climate data sources and processing methods.

(3) Please explain the reasons for the selection of these landscape indicators in the research methodology section.

(4) The water area at the three times in Table 3 is not consistent with the size of the water area in Fig. 3. Similarly, the area of water converted to bareland in Fig. 5 is concentrated in the north and the south, but this is deviation from the distribution of water and bareland in Fig. 3. Please check the accuracy of the data throughout the paper.

(5) The authors use CA-Markov for future land use (including forests) projections, but there is a lack of description of the parameter settings for this model. In addition, how is it possible to predict land use data (2018-2048) for the next 30 years with 20 years (1998-2018) of historical land use data. Based on the characteristics of the model, it is difficult to complete the prediction of this data, please explain in detail in the research methodology.

(6) Although the authors consider human activities and climate change to be important factors influencing land use change, the authors only quantitatively analysed the impact of climate change on land use change (Table 11), while the impact of human activities (urbanisation, agricultural activities) lacks quantitative analysis.

Minor editing of English language required

Author Response

Reviewer-2

(1) It is suggested that in the introduction section, the limitations of the current research as well as the innovations of this study should be added.

Ans: A short passage has been added in the introduction section to highlight the need of this research, and significance of this research.

(2) The authors only introduce the remote sensing data, and lack of explanation on the climate data sources and processing methods.

Ans: Added. Please see the methodology section of MS.

(3) Please explain the reasons for the selection of these landscape indicators in the research methodology section.

Ans: In this study area, the FRAGSTATES was applied for computed landscape metrics. Landscape metrics were used to characterize the spatial fragmentation and heterogeneity for 1998, 2008 and 2018 covering three decades. FRAGSTATS is a spatial pattern analysis program was implemented by decision maker, forest manager and ecologists to analyze landscape fragmentation or describing characteristics of landscape, elements of those landscapes (KeleÅŸ et al., 2008). The software FRAGSTATS version 4.2.1 was used to extract the landscape metrics from each classified map of 1998, 2008 and 2018. In this study, total twelve landscape metrics including:

- class level: class area (CA), total landscape area (TLA), number of patch (NP), mean patch size (MPS), patch size coefficients of Varian (PSCOV), mean shape index (MSI), edge density (ED), mean nearest neighbor distan (MNN), Interspersion Juxtaposition Index (IJI),

- landscape level:total landscape area (TLA), number of patch (NP), mean patch size (MPS), mean shape index (MSI), mean nearest neighbor distan (MNN), Interspersion Juxtaposition Index (IJI), Mean Proximity Index (MPI), Shannon’s Diversity Index(SDI), Shannon’s Evenness Index (SHEI).

A brief definition of landscape metrics used in study was given in Table.

Landscape metrics

Description

Range

Unit

Class area (CA)

Total amount of class area in the landscape

CA > 0, without limit

Hectares

Number of patches (NP)

Number of patches of landscape classes

NP ≥ 1, without limit

None

Mean Patch Size (MPS)

The average area of patch corresponding to the area

Hectares

Patch Size Coefficient of Varian (PSCOV)

Coefficient of variation of patches

None

Hectares

Mean Shape Index (MSI)

sum of each patch's perimeter divided by the square root of patch area (in hectares) for each class

SHAPE ≥ 1, without limit

None

Edge Density (ED)

Amount of edge relative to the landscape area

ED ≥ 0, without limit

Meters per hectare

Mean Nearest NeighbourDistan

(MNN)

The mean nearest neighbor distance is the average of these distances (metres) for individual classes at the class level and the mean of the class nearest neighbor distances at the landscape level

None

Meters

Interspersion Juxtaposition Index(IJI)

Interspersion and juxtaposition index measure the juxtaposition of a focal patch class with all other classes

0 < IJI ≤ 100

Percentage

Mean Proximity Index(MPI)

Average proximity index for all patches in a class

PROXE ≥ 0

None

Shannon’sDiversity Index(SDI)

Shannon’s diversity index is amount of patch per individual

SDI ≥ 0, without limit

Information

Shannon’s Evenness Index(SHEI)

Shannon’s evenness index is the observed level of diversity divided by the maximum possible diversity for a given patch richness

0 ≤ SHEI ≤ 1

None

(4) The water area at the three times in Table 3 is not consistent with the size of the water area in Fig. 3. Similarly, the area of water converted to bareland in Fig. 5 is concentrated in the north and the south, but this is deviation from the distribution of water and bareland in Fig. 3. Please check the accuracy of the data throughout the paper.

Ans: Revised as suggested.

(5) The authors use CA-Markov for future land use (including forests) projections, but there is a lack of description of the parameter settings for this model. In addition, how is it possible to predict land use data (2018-2048) for the next 30 years with 20 years (1998-2018) of historical land use data. Based on the characteristics of the model, it is difficult to complete the prediction of this data, please explain in detail in the research methodology.

Ans: Added. Please see the methodology section of MS.

(6) Although the authors consider human activities and climate change to be important factors influencing land use change, the authors only quantitatively analysed the impact of climate change on land use change (Table 11), while the impact of human activities (urbanisation, agricultural activities) lacks quantitative analysis.

Ans: Thanks for highlight this point. We have addressed the problem for detail Please see the methodology section of MS.

Round 2

Reviewer 2 Report

No comments.

No comments.